# Effectiveness of the Boston Brace in the Treatment of Paediatric Scoliosis: A Longitudinal Study from 2010–2020 in a National Spinal Centre

**DOI:** 10.3390/healthcare11101491

**Published:** 2023-05-20

**Authors:** Athanasios I. Tsirikos, Rachel Adam, Kirsty Sutters, Maureen Fernandes, Silvia García-Martínez

**Affiliations:** Scottish National Spine Deformity Centre, Royal Hospital for Children and Young People, Edinburgh EH16 4TJ, UK

**Keywords:** scoliosis, idiopathic scoliosis, adolescent idiopathic scoliosis, bracing, brace treatment, outcomes, early-onset scoliosis, non-idiopathic scoliosis, SRS-22 questionnaire

## Abstract

Bracing can reduce curve progression in order to prevent or delay scoliosis surgery in growing children. Brace treatment is effective in adolescent idiopathic scoliosis (AIS), but there is less evidence of its efficacy in early-onset or non-idiopathic scoliosis. We assessed the outcome of bracing at the end point of treatment, including the patients’ perception of clinical results. We reviewed 480 patients treated using Boston brace from 2010–2020 (70% female); 249 patients completed bracing (52%) and 118 patients (47.4%) did not require surgery, with 83% having idiopathic scoliosis. Brace success was considered scoliosis below 50° at the end of bracing, with the patient skeletally mature. A total of 131 patients required scoliosis surgery after bracing (64% had idiopathic scoliosis; adolescents 57% and juveniles 43%). All patients had a minimum two-year follow-up after bracing or after scoliosis correction, with the quality of life assessment questionnaires. A total of 98 out of 182 patients with idiopathic scoliosis did not require surgery (54%). Thoracic scoliosis improved with bracing by a mean of 3.4° and thoracolumbar/lumbar scoliosis by a mean of 6.8°. A total of 85 patients with AIS (64%) but only 9 patients with JIS (20%) did not need surgery. In the AIS group, 97 patients had scoliosis of 20–40°; 71 of these patients (73.2%) did not require scoliosis correction at the end of bracing. In total, 84 patients with idiopathic scoliosis had surgery at a mean of 14 years (surgery was delayed by a mean of 3.2 years). In total, 20 of 67 patients with non-idiopathic scoliosis did not need surgery (30%). Thoracic scoliosis improved with bracing by a mean of 8.4° and thoracolumbar/lumbar scoliosis by a mean of 0.8°. A total of 47 patients with non-idiopathic scoliosis required surgery at a mean of 13.1 years (surgery was delayed by a mean of 5.2 years). Multivariate regression analysis showed that idiopathic scoliosis, AIS, closed triradiate cartilage, post-menarche status, higher Risser grade and smaller scoliosis angle at initial presentation predicted brace success. Patients reported good function and self-image, reduced pain and high satisfaction after treatment in both the bracing-only and the bracing followed by surgery groups.

## 1. Introduction

Scoliosis is characterised by a lateral curvature of the spine greater than 10° associated, with vertebral rotation and rib cage deformity towards the convexity of the curve. Sagittal imbalance occurs and is typically presented as thoracic hypokyphosis. Idiopathic scoliosis (IS) is the most common type, which includes up to 80% of the structural coronal curvatures [1]. Within the IS group, adolescent idiopathic scoliosis (AIS) has the highest prevalence, accounting for 3% of patients between the ages of 10 and 18 years [2]. Curves of approximately 50° at skeletal maturity carry an increased risk of slow deterioration into adult life at a rate of around 1° per year and, therefore, indicate the need for surgical correction [3,4]. Weinstein et al. [5] looked into the natural history of untreated idiopathic scoliosis and reported that 22% of patients with thoracic curves over 80° had shortness of breath that affected activities of daily living, while chronic back pain occurred in 61% of patients, with 68% of them having low or moderate intensity.

Brace treatment is recommended for curves of 20–40° in skeletally immature patients with Risser grade 0–1 to prevent scoliosis progression during a period of spinal growth [6]. Following these indications, previous studies on bracing in AIS have documented its effectiveness in decreasing curve deterioration with bracing in adolescent idiopathic scoliosis trial (BRAIST), providing conclusive evidence that the progression of high-risk curves to the threshold for surgery is significantly reduced [7]. The use of a brace has no documented impact on curves over 40°; in this group of patients, bracing can potentially slow down rather than prevent deterioration of the curve and delay scoliosis correction while avoiding the need for repetitive growth-friendly surgeries, such as growing rods.

Scoliosis can also develop in patients with congenital, neuromuscular or syndromic conditions, as well as in association with intraspinal abnormalities, congenital cardiac disease, open heart surgery or a spinal tumour [8,9,10]. The curves in many of these patients resemble idiopathic scoliosis; however, the impact of brace treatment in controlling the deformity is less investigated [11,12].

The purpose of this study was to analyse the data of all patients who underwent bracing treatment at the National Spinal Centre and assess the factors that can predict a successful outcome. We evaluated a group of patients in terms of underlying diagnosis, type and degree of scoliosis before and after the treatment, length of bracing and the final results at the end of bracing for patients who did not require surgery or at the end of postoperative follow-ups for patients who completed bracing but needed scoliosis correction, including patient-reported outcomes.

## 2. Material and Methods

We performed a retrospective analysis of prospectively collected data in all patients with scoliosis who underwent bracing using the Boston brace from 2010–2020 at the National Spinal Centre. These patients were divided into those with idiopathic scoliosis (IS) and those with non-idiopathic scoliosis (non-IS). The group of IS patients was further subdivided into adolescent (AIS), juvenile (JIS) and infantile idiopathic scoliosis (IIS). All patients with IS had a spinal MRI that excluded intraspinal abnormalities. Patients with non-IS scoliosis were subdivided into neuromuscular (NM), congenital or syndromic scoliosis, and scoliosis secondary to intraspinal anomalies (Chiari I malformation and/or syringomyelia), congenital cardiac disease/open cardiac surgery or an intraspinal tumour (Table 1). Patients with IS and non-IS were analysed separately for all categories of radiographic and clinical data.

Our indication for bracing included patients with an IS of 20–40° at initial presentation who had significant remaining spinal growth (Risser grade 0–1). In this group, the aim was to prevent scoliosis progression and avoid the need for surgery. Brace treatment was also used in young children with IS over 40° in order to delay curve deterioration and the need for spinal fusion as an alternative to growing rod procedures. Similarly, we applied bracing in young patients with non-IS who developed flexible curves in order to postpone the need for scoliosis correction. We recommended 20 h of brace wear per day for all groups of patients. None of the patients included in this study required growth-friendly surgery, and spinal fusion was performed after the age of 10 years. This allowed the most significant stages of alveolar and chest growth to occur [13].

The primary outcome of this study was defined according to the criteria used in the BRAIST study [7]. *Treatment failure* was considered when scoliosis at the completion of bracing had progressed to 50° or more, which indicated the need for scoliosis correction. *Treatment success* was when the curve was below 50° at the end of brace treatment, with the patient skeletally mature (Risser grade 5 and Sanders digital maturity stage 8) [14]. All patients in both the *treatment failure* and *treatment success* groups were followed to skeletal maturity, apart from six patients with early-onset scoliosis. Four of these patients had IIS that was corrected during bracing. In addition, 2 patients had scoliosis associated with Chiari I malformation and underwent foramen magnum decompression during bracing. Scoliosis gradually improved, which allowed bracing to be discontinued.

The effectiveness of bracing and the final outcome of treatment were assessed on the basis of scoliosis angle measurements that were performed on serial spinal radiographs taken at the initial clinical visit before bracing was initiated, in the first brace assessment in the clinic, and at the last clinical follow-up, which was at minimum 2 years after completion of brace treatment. In addition, for patients who underwent scoliosis correction, scoliosis measurements were taken before surgery and at minimum 2 years later in the postoperative follow-up.

The quality of life assessment data was used to evaluate the patients’ perception of clinical outcomes using a non-validated brace questionnaire, as well as the Scoliosis Research Society 22r questionnaire (SRS-22r). Patients at the end of brace treatment who did not require surgery completed both the brace questionnaire and the SRS-22r questionnaire. Patients who underwent scoliosis correction at the end of bracing completed the SRS-22r questionnaire before the surgery, as well as at 6, 12 and 24 months after surgery. All patients with IS and non-IS who were cognitively able completed the brace questionnaire. All patients who were cognitively able and had normal neurological functions completed the SRS-22r questionnaire in the IS and non-IS groups.

### Statistical Analysis

Data were analysed using IBM SPSS v. 27.0 (Armonk, NY, USA). The Shapiro–Wilk test was used to assess data normality. Independent samples *t*-test compared continuous parametric data between the groups. Two-tailed *p*-values were reported, with significance at *p* < 0.05. Multivariate logistic regression analysis was performed to identify the risk factors that could independently predict the success of brace treatment. This included calculation of the odds ratios (OR) and their 95% confidence intervals (CI).

## 3. Results

During the study period (2010–2020), 480 consecutive patients were treated with a Boston brace (336 female—70%; 144 male—30%). A total of 249 patients completed brace treatment (52%), including 176 females (70.7%) and 73 males (29.3%). Of the 249 patients, 118 patients (47.4%) completed treatment and did not require scoliosis surgery. In total, 98 of the 118 patients (83%) who completed brace treatment and did not require scoliosis surgery had IS and 20 patients (17%) had non-IS. A total of 131 of the 249 patients (52.6%) who underwent bracing required scoliosis surgery. Of the 131 patients, 84 patients (64%) who required scoliosis surgery at the end of brace treatment had IS, and 47 patients (36%) had non-IS. Of the 84 patients with IS who required scoliosis surgery at the end of bracing, 48 patients had AIS (57%) and 36 had JIS (43%).

### 3.1. Patients with IS

A total of 321 of the total group of 480 patients (67%) who underwent bracing had IS. Of the 321 patients, 182 patients with IS (57%) completed the brace treatment. Of these 182 patients, 98 patients did not require scoliosis surgery (54%) and 84 patients underwent scoliosis correction at the completion of bracing (46%) (Table 2 and Table 3). In total, 133 patients (73%) had AIS, 45 had JIS (25%) and 4 had IIS (2%).

Among 133 patients with AIS who completed treatment, 85 patients (64%) did not require scoliosis surgery and 48 patients (36%) underwent scoliosis correction. Ninety-seven patients who completed treatment had scoliosis between 20–40° (Table 4). Among these 97 patients, 71 patients (73.2%) did not require scoliosis surgery and 26 patients (26.8%) underwent scoliosis correction at the end of bracing.

Among 45 patients with JIS who completed treatment, 9 patients (20%) did not require scoliosis surgery and 36 patients (80%) underwent scoliosis correction. Four patients with IIS who completed treatment did not require scoliosis surgery.

#### 3.1.1. Patients with IS at the End of Bracing/No Need for Scoliosis Surgery (98 Patients; 85 AIS, 9 JIS, 4 IIS)

In this group, 19 patients (19%) had thoracic, 50 patients (51%) had thoracic and lumbar, 5 patients (5%) had double thoracic and 24 patients (25%) had thoracolumbar/lumbar scoliosis (Table 2). Mean thoracic scoliosis before bracing was 28.7° (range: 13–56°); in the first brace, it was 18.6° (range: 0–38°). At the last follow-up, mean thoracic scoliosis was 25.3° (range: 0–42°) with a mean improvement of 3.4° (*p* = 0.04) during treatment. Mean thoracolumbar/lumbar scoliosis before bracing was 26.5° (range: 13–50°) and in the first brace 16.9° (range: 0–36°). At the last follow-up, mean thoracolumbar/lumbar scoliosis was 19.8° (range: 0–40°) with a mean improvement of 6.8° (*p* = 0.03) during treatment. The triradiate cartilage at the start of bracing was open in 41 patients (46%) and closed in 48 patients (54%). Mean Risser grade at the start of bracing was 0.4 (range: 0–2); at the end of bracing, it was 4.7 (range: 0–5). Mean age of patients was 11.9 years (range: 1.9–15.9) at the start of bracing and 15.7 years (range: 6.4–18.4) at the end of bracing. Mean length of bracing was 3.9 years (range: 0.5–14.4). Mean follow-up beyond the end of bracing was 2.2 years (range: 2–2.5).

#### 3.1.2. Patients with IS at the End of Bracing Who Underwent Scoliosis Surgery (84 Patients; 48 AIS, 36 JIS)

In this group, 19 patients (23%) had thoracic, 52 patients (63%) had thoracic and lumbar, 7 patients (8%) had double thoracic and 5 patients (6%) had thoracolumbar/lumbar scoliosis (Table 3). Mean thoracic scoliosis before bracing was 40.8° (range: 32–70°); in the first brace, it was 30.7° (range: 8–65°). At the end of bracing, mean thoracic scoliosis was 56.3° (range: 47–97°); before scoliosis surgery, it was 59.8° (range: 48–97°). At the last follow-up, mean thoracic scoliosis was 22.4° (range: 4–48°), indicating a mean improvement of 18.4° (*p* = 0.008) during treatment from pre-brace to postoperative follow-up. Mean thoracolumbar/lumbar scoliosis before bracing was 33.2° (range: 20–54°); in the first brace, it was 26.4° (range: 9–40°). At the end of bracing, mean thoracolumbar/lumbar scoliosis was 55° (range: 42–69°); before scoliosis surgery, it was 57.9° (range: 48–69°). At the last follow-up, mean thoracolumbar/lumbar scoliosis was 18.2° (range: 2–35°), indicating a mean improvement of 15° (*p* = 0.009) during treatment from pre-brace to postoperative follow-up. The triradiate cartilage at the start of bracing was open in 67 patients (81%) and closed in 16 patients (19%). Mean Risser grade at the start of bracing was 0.1 (range: 0–2); at the end of bracing, it was 3.5 (range: 0–5). Mean age of patients was 10.5 years (range: 3.1–15.8) at the start of bracing and 13.7 years (range: 9.8–17.7) at the end of bracing. Mean age at scoliosis surgery was 14 years (range: 10.2–18.4). Mean length of bracing was 3.2 years (range: 0.6–9.4). All patients had a minimum 2-year follow-up after scoliosis surgery (mean: 3.2 years; range: 2–4.8).

### 3.2. Patients with Non-IS

In total, 159 of the total group of 480 patients (33%) who underwent bracing had non-IS. Of the 159 patients, 67 patients (42%) completed brace treatment. The underlying diagnosis included 10 patients with neuromuscular (NM), 24 patients with syndromic, 14 patients with congenital scoliosis where bracing was used to treat structural compensatory curves, and 19 patients with scoliosis secondary to Chiari I malformation/syringomyelia, congenital cardiac disease/open cardiac surgery or an intraspinal tumour (Table 1). Of the 67 patients who completed treatment, 20 patients did not require scoliosis surgery (30%) and 47 patients underwent scoliosis correction (70%) (Table 5).

#### 3.2.1. Patients with Non-IS at the End of Bracing/No Need for Scoliosis Surgery (20 Patients; 5 with Congenital Scoliosis, 5 with NM Scoliosis, 6 with Syndromic Scoliosis, 4 with Secondary Scoliosis)

In this group, 3 patients (15%) had thoracic, 8 patients (40%) had thoracic and lumbar and 9 patients (45%) had thoracolumbar/lumbar scoliosis. Mean thoracic scoliosis before bracing was 30.3° (range: 21–43°); in the first brace, it was 21.6° (range: 12–31°). At the last follow-up, mean thoracic scoliosis was 21.9° (range: 0–44°) with a mean improvement of 8.4° (*p* = 0.02) during treatment. Mean thoracolumbar/lumbar scoliosis before bracing was 29° (range: 21–65°); in the first brace, it was 21.8° (range: 7–33°). At the last follow-up, mean thoracolumbar/lumbar scoliosis was 28.2° (range: 0–43°) with a mean improvement of 0.8° (*p* = 0.12) during treatment. The triradiate cartilage at the start of bracing was open in 15 patients (75%) and closed in 5 patients (25%). Mean Risser grade at the start of bracing was 0.5 (range: 0–2); at the end of bracing, it was 4.8 (range: 0–5). Mean age of patients was 9.6 years (range: 1.7–14.8) at the start of bracing and 15.8 years (range: 4.8–18.6) at the end of bracing. Mean length of bracing was 5.9 years (range: 1.1–13.1). Mean follow-up beyond the end of bracing was 2.3 years (range: 2–2.7).

#### 3.2.2. Patients with Non-IS at the End of Bracing Who Underwent Scoliosis Surgery (47 Patients; 9 with Congenital Scoliosis, 3 with NM Scoliosis, 21 with Syndromic Scoliosis, 14 with Secondary Scoliosis)

In this group, 16 patients (34%) had thoracic, 25 patients (53%) had thoracic and lumbar, 1 patient (2%) had double thoracic, and 5 patients (11%) had thoracolumbar/lumbar scoliosis (Table 5). Mean thoracic scoliosis before bracing was 41° (range: 20–78°); in the first brace, it was 31.6° (range: 14–54°). At the end of bracing, mean thoracic scoliosis was 66.3° (range: 48–117°); before scoliosis surgery, it was 69.7° (range: 52–117°). At the last follow-up, mean thoracic scoliosis was 32.2° (range: 17–63°), indicating a mean improvement of 8.8° (*p* = 0.05) during treatment from pre-brace to postoperative follow-up. Mean thoracolumbar/lumbar scoliosis before bracing was 34.7° (range: 14–58°); in the first brace, it was 24.5° (range: 10–35°). At the end of bracing, mean thoracolumbar/lumbar scoliosis was 60.8° (range: 49–85°); before scoliosis surgery, it was 63.3° (range: 51–85°). At the last follow-up, mean thoracolumbar/lumbar scoliosis was 22° (range: 8–50°), indicating a mean improvement of 12.7° (*p* = 0.01) during treatment from pre-brace to postoperative follow-up. The triradiate cartilage at the start of bracing was open in 46 patients (98%) and closed in 1 patient (2%). Mean Risser grade at the start of bracing was 0.1 (range: 0–2); at the end of bracing, it was 3.1 (range: 0–5). Mean age of patients was 7.2 years (range: 1.6–14.2) at the start of bracing and 12.8 years (range: 9.6–16.7) at the end of bracing. Mean age at scoliosis surgery was 13.1 years (range: 10.3–16.8). Mean length of bracing was 5.2 years (range: 0.5–11.8). All patients had a minimum 2-year follow-up after scoliosis surgery (mean: 4.3 years; range: 2.4–5).

### 3.3. Multivariate Logistic Regression Analysis

We applied multivariate regression analysis models to assess factors that could predict the success of brace treatment. This analysis demonstrated that the diagnosis of IS as opposed to non-IS (OR = 3.982, 95%CI:2.771–4.956; *p* = 0.008), the diagnosis of AIS compared to JIS (OR = 2.365, 95%CI:1.917–3.867; *p* = 0.01), a high Risser grade (OR = 1.518, 95%CI:0.872–2.914; *p* = 0.04), a closed as opposed to open triradiate cartilage (OR = 1.719, 95%CI:1.165–3.475; *p* = 0.02), a post-menarche status (OR = 1.894, 95%CI:1.243–3.576; *p* = 0.03) and a reduced scoliosis angle at the start of treatment (OR = 1.684, 95%CI:1.153–2.657; *p* = 0.03) predicted a successful outcome of bracing (Table 6).

### 3.4. Patient Reported Outcomes (End of Bracing and after Scoliosis Surgery)

Among 118 patients with IS and non-IS who underwent bracing treatment and did not require surgery, 109 patients completed the brace questionnaire. The results showed a high level of satisfaction (Table 7). A total of 95 patients with IS who underwent bracing treatment and did not require surgery completed the SRS-22r questionnaire, reporting a mean satisfaction rate of 4.4 (Figure 1). Fourteen patients with non-IS who had bracing treatment and did not require surgery completed the SRS-22r questionnaire, reporting a mean satisfaction rate of 4.9 (Figure 2).

All 84 patients with IS (AIS/JIS) who underwent bracing and surgery completed the SRS-22r questionnaire and reported improvement in the total and domain scores between preoperative and minimum 2-year postoperative with a mean satisfaction rate of 4.78 (Figure 3). In total, 22 patients with non-IS who underwent bracing and had scoliosis correction completed the SRS-22r questionnaire reporting improvement in total and domain scores with a mean satisfaction rate of 4.83 (Figure 4).

## 4. Discussion

Bracing is the most commonly used conservative treatment modality in AIS. In adolescents with IS who are considered to be at a high risk of curve progression, the weight of evidence is in favour of bracing over observation. The target group included patients with skeletal immaturity (Risser grade 0–2) and scoliosis between 20 and 40°. The efficacy of brace treatment has been demonstrated in BRAIST. This study provided high-quality data for the application of full-time rigid bracing that resulted in a treatment success rate of 72% in preventing scoliosis progression to 50°, which is a common indication for surgery, compared to 48% when observation alone was applied [7]. Bracing in patients with curves greater than 40° and in patients with non-IS is less effective and has not been clearly documented [15].

Bracing concepts are distinguished according to the wearing time in full-time, part-time and night-time; they are also differentiated into rigid and soft braces. There is currently insufficient evidence on whether there is a difference in effectiveness between brace types, with the majority of bracing studies having a significant risk of bias. This is partly due to the difficulty of performing prospective controlled trials in patients with this condition. A recent systematic review and meta-analysis did not identify a significant difference in the outcome of treatment between full-time and night-time braces [16]. In contrast, rigid braces have a higher success rate compared to soft braces.

In this study, we reviewed the results of the use of the Boston brace in a large population of children and adolescents with different types of scoliosis over a 10-year period. Of the 249 patients who completed scoliosis treatment, 47.4% did not require scoliosis correction. In the group of IS, 54% of patients completed bracing and had no scoliosis surgery. Brace treatment prevented the need for scoliosis correction in 73.2% of adolescents with IS of 20–40°. These results are comparable to those presented in BRAIST. In addition, bracing avoided scoliosis surgery in 64% of all AIS patients, which included curves over 40° at the start of brace treatment.

In the JIS group, bracing demonstrated less effectiveness compared to patients with AIS, as only 20% of patients completed brace treatment and did not require surgery. However, scoliosis correction in JIS patients was delayed by a mean of 5 years, which preserved spinal growth and chest development while preventing the need for repeated growth-friendly procedures. The requirement for scoliosis correction is recognised in this group of patients due to the early-onset nature of their scoliosis, which is associated with a high risk of rapid progression and often the need for surgery in the teenage years.

Similarly, 30% of patients with non-IS reached the end of bracing, with no need for scoliosis correction. In the remaining patients with complex non-IS, scoliosis surgery was delayed by a mean of 5.2 years, which avoided repetitive growing rod lengthenings. These are patients with a known underlying aetiology for their scoliosis (neuromuscular, congenital, syndromic, secondary to intraspinal anomalies, cardiac disease or a spinal tumour), which is characterised by early curve onset, rapid deterioration and the need for surgical treatment, commonly in early adolescence. In this group, bracing allowed for delaying scoliosis surgery for a number of years and preserved spinal and chest growth. This is critical for the patients’ general well-being and long-term survivorship, especially in the presence of their underlying medical conditions and associated comorbidities.

Determining the success of bracing can be a challenge. A benchmark of the effectiveness of brace treatment can be defined as scoliosis progression of less than 5–10° by skeletal maturity or preventing the curve from reaching 50° at the end of spinal growth. The latter is the target that was used for the success of brace treatment in BRAIST and in our study. In the present study, we did not monitor patients’ compliance with bracing using brace-fitted sensors. The SRS guidelines support this practice and suggest that the efficacy of brace treatment can be evaluated on the actual success rate in controlling scoliosis and preventing the need for surgery, regardless of patient compliance [17].

The degree of scoliosis angle, Risser grade and vertebral rotation have been shown to have a prognostic effect on bracing [18,19]. In a study of 93 AIS patients treated with the Rigo System Cheneau brace, Ovadia et al. [20] used multivariable logistic regression models and reported that a high Risser score, low scoliosis angle and low angle of thoracic rotation pre-treatment were associated with brace treatment success. Following the same statistical approach, our study demonstrated that brace success for scoliosis at the end of growth below 50°, which prevented the necessity for surgery, was higher in IS compared to non-IS, AIS compared to JIS, closed compared to open triradiate cartilage, post-menarche status, with a high Risser grade and a lesser scoliosis angle at the beginning of treatment. These are useful data that can inform brace care in patients with paediatric and adolescent-onset scoliosis.

Growing age, especially adolescence, is a transitional period of physical and psychological development that is commonly associated with emotional instability and fluctuation in mood. This can make compliance with wearing an externally visible brace challenging as this is affected by peer pressure, and the effectiveness of treatment requires close collaboration of the patient and his/her family with the surgical and orthotic teams in order to provide maximum support during the length of treatment and foster bracing success. The outcome of any treatment can be defined by the patient’s perception of the results, which is the reason why we included in our analysis a brace questionnaire and the SRS-22r outcome measure. We recorded high patient satisfaction at the latest follow-up among the groups that completed bracing, with or without the need for scoliosis surgery. This was associated with normal function, good self-image and reduced pain at the end of the scoliosis treatment (Table 7, Figure 1, Figure 2, Figure 3 and Figure 4).

This study has some limitations. It is an ongoing study, and we plan to update our results as more patients complete their scoliosis treatment. In addition, we are conducting follow-ups of these patients in their adult lives to determine whether there is evidence of curve progression occurring beyond skeletal maturity and potentially requiring surgical treatment. We are also keeping under close review the six patients with early-onset scoliosis who completed bracing but still carry a risk of scoliosis recurrence due to the remaining growth. Finally, we do not have groups of patients with non-IS similar to those treated in a brace to use as control and compare the natural history of scoliosis under observation, which would more clearly illustrate the efficacy and value of bracing.

In conclusion, in this study, bracing using the Boston brace avoided scoliosis surgery in nearly three-fourths of patients with AIS of 20–40° and 54% of all patients with IS. We found brace treatment useful in delaying scoliosis correction among patients with JIS or non-IS, as this postponed surgery for a mean of 5 and 5.2 years, respectively, while preventing the need for repeat growth-friendly procedures that carry increased risks of major complications and recurrent morbidity. At the end of treatment, patients reported normal function, good self-image, less pain and high satisfaction in both the bracing-only and bracing followed by the surgery groups.

## Figures and Tables

**Figure 1 healthcare-11-01491-f001:**
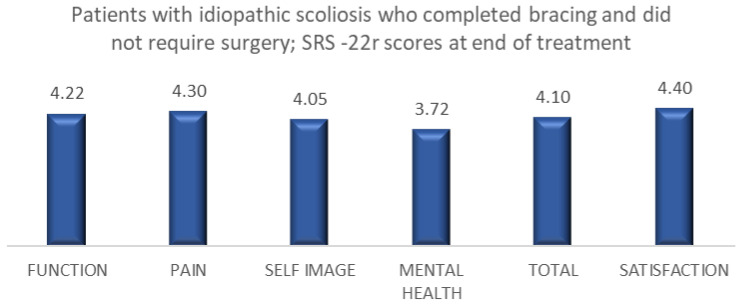
SRS-22r questionnaire results among patients with IS who completed brace treatment and did not require scoliosis surgery (95 of 98 patients completed the questionnaire).

**Figure 2 healthcare-11-01491-f002:**
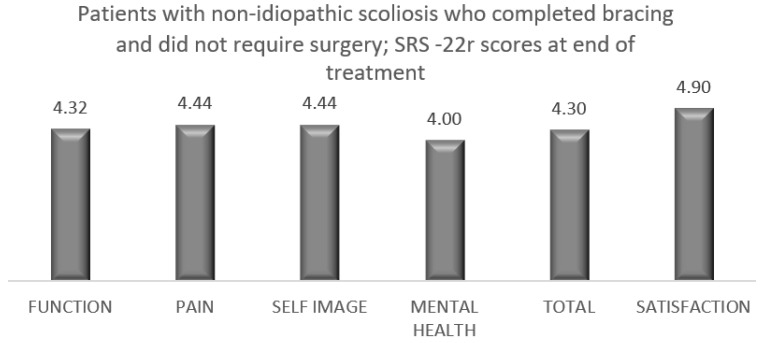
SRS-22r questionnaire results among patients with non-IS who completed brace treatment and did not require scoliosis surgery (14 of 20 patients completed the questionnaire).

**Figure 3 healthcare-11-01491-f003:**
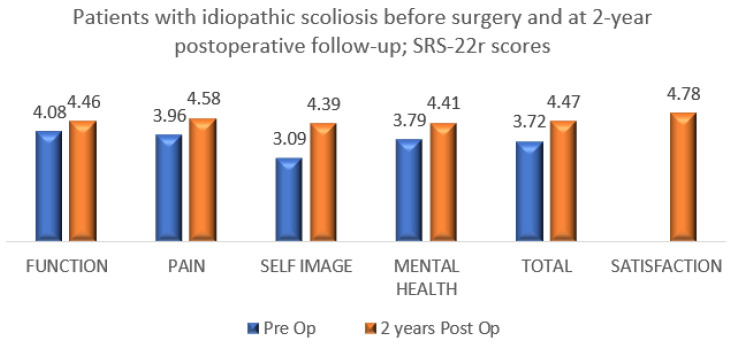
SRS-22r questionnaire results among 84 patients with IS who completed bracing and underwent scoliosis surgery (preoperative compared to 2-year postoperative follow-up).

**Figure 4 healthcare-11-01491-f004:**
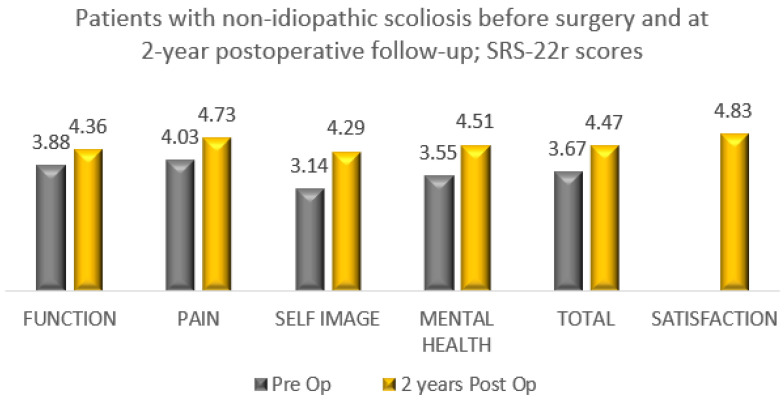
SRS-22r questionnaire results among patients with non-IS who completed bracing and underwent scoliosis surgery (preoperative compared to 2-year postoperative follow-up). A total of 22 of 47 patients completed the SRS-22r questionnaire (47%).

**Table 1 healthcare-11-01491-t001:** Underlying diagnosis in the group of patients with non-IS who completed treatment and underwent bracing-only or bracing followed by surgical correction.

Diagnosis	Neuromuscular (10 Patients)	Syndromic (24 Patients)	Secondary (19 Patients)
Hemiplegic cerebral palsy	5	-	-
Charcot–Marie–Tooth disease	3	-	-
Congenital hypotonia	2	-	-
Scoliosis with Chiari I malformation and syringomyelia	-	-	7
Scoliosis with syringomyelia	-	-	5
Scoliosis with congenital cardiac disease	-	-	5
Scoliosis with intraspinal tumour	-	-	2
Chromosome abnormality	-	10	-
Marfan syndrome	-	4	-
Neurofibromatosis type-1	-	2	-
Pierre Robin syndrome	-	2	-
Prader–Willi syndrome	-	2	-
Ehlers–Danlos syndrome	-	1	-
Hurler syndrome	-	1	-
Cleidocranial dysostosis	-	1	-
Carey–Fineman–Ziter syndrome	-	1	-

**Table 2 healthcare-11-01491-t002:** Patients with IS who completed brace treatment and did not need scoliosis surgery.

Data	AIS (85 Patients)	JIS (9 Patients)	IIS (4 Patients)
Type of scoliosis	TH: 10; TH and L: 41; double TH: 4; TL/L: 21	TH: 2; TH and L: 6; double TH: 1	TH: 1; TL/L:3
TH scoliosis pre-brace	28° (range: 16–43°)	32° (range: 13–56°)	56°
TH scoliosis in 1st brace	19° (range: 0–38°)	16° (range: 0–24°)	24°
TH scoliosis at last follow-up	26° (range: 0–42°)	20° (range: 0–40°)	17°
Mean TH scoliosis improvement	7°	5°	41°
TL/L scoliosis pre-brace	28° (range: 13–44°)	21° (range: 20–35°)	43° (range: 36–50°)
TL/L scoliosis in 1st brace	17° (range: 0–34°)	11° (range: 0–22°)	26° (range: 19–36°)
TL/L scoliosis at last follow-up	21° (range: 0–40°)	16° (range: 0–20°)	2° (range: 0–5°)
Mean TL/L scoliosis improvement	7°	5°	41°
Triradiate cartilage	Open: 28 Closed: 48	Open: 9	Open: 4
Risser grade (start of bracing)	0.4 (range: 0–2)	0	0
Risser grade (end of bracing)	4.8 (range: 0–5)	4.7 (range: 3–5)	0 (range: 0–1)
Age (start of bracing); years	12.9 (range: 10.1–15.9)	7.2 (range: 3.3–10)	2.3 (range: 1.9–2.4)
Age (end of bracing); years	16.1 (range: 12.8–18.4)	15.5 (range: 13.6–18.1)	8.8 (range: 6.4–12.2)
Years in brace treatment	3.2 (range: 0.5–9.1)	8.4 (range: 4.1–14.4)	6.4 (range: 3.8–9.6)
Follow-up (after bracing); years	2.2 (range: 2–2.5)	2.1 (range: 2–2.3)	2 (range: 2–2.1)

TH: thoracic; TL: thoracolumbar; L: lumbar.

**Table 3 healthcare-11-01491-t003:** Patients with IS who completed brace treatment and underwent scoliosis surgery.

Data	AIS (48 Patients)	JIS (36 Patients)
Type of scoliosis	TH: 9; TH and L: 31; double TH: 3; TL/L: 5	TH: 10; TH and L: 21; double TH: 4
TH scoliosis pre-brace	35.4° (range: 15–64°)	41° (range: 22–70°)
TH scoliosis in 1st brace	27.7° (range: 8–51°)	27.7° (range: 10–70°)
TH scoliosis at end of bracing	48.9° (range: 17–84°)	59.8° (range: 11–97°)
TH scoliosis preoperatively	50.4° (range: 17–84°)	64.3° (range: 46–97°)
TH scoliosis at last follow-up	18.1° (range: 4–41°)	24.9° (range: 4–50°)
Mean TH scoliosis improvement	17.3°	16°
TL/L scoliosis pre-brace	32.4° (range: 18–51°)	33.6° (range: 17–54°)
TL/L scoliosis in 1st brace	24.4° (range: 14–51°)	24.4° (range: 9–46°)
TL/L scoliosis at end of bracing	45.1° (range: 22–66°)	47.8° (range: 12–69°)
TL/L scoliosis preoperatively	48.8° (range: 28–66°)	54.4° (range: 31–69°)
TL/L scoliosis at last follow-up	17.6° (range: 2–45°)	19.1° (range: 4–29°)
Mean TL/L scoliosis improvement	14.8°	14°
Triradiate cartilage	Open: 32 Closed: 16	Open: 36
Risser grade (start of bracing)	0.1 (range: 0–2)	0
Risser grade (end of bracing)	4 (range: 4–5)	3.5 (range: 0–5)
Age (start of bracing); years	12.5 (range: 10.2–15.8)	7.8 (range: 3.1–10)
Age (end of bracing); years	14.6 (range: 11.2–17.7)	12.5 (range: 9.8–16.8)
Age at scoliosis surgery; years	14.9 (range: 11.6–18.4)	12.8 (range: 10.2–16.8)
Years in brace treatment	2.4 (range: 0.6–5.1)	5 (range: 1.6–9.4)
Follow-up (post-surgery); years	2.8 (range: 2–3.5)	3.5 (range: 2.6–4.8)

TH: thoracic; TL: thoracolumbar; L: lumbar.

**Table 4 healthcare-11-01491-t004:** Patients with AIS with original curves between 20–40° who completed brace treatment (with and without surgery).

Data	AIS without Surgery (71 Patients)	AIS with Surgery (26 Patients)
Type of scoliosis	TH: 9; TH and L: 37; double TH: 4; TL/L: 21	TH: 3; TH and L: 15; double TH: 3; TL/L:5
TH scoliosis pre-brace	26.3° (range: 16–40°)	29.1° (range: 15–39°)
TH scoliosis in 1st brace	18.2° (range: 0–38°)	22.8° (range: 8–36°)
TH scoliosis at end of bracing	25.4° (range: 0–45°)	44° (range: 17–65°)
TH scoliosis preoperatively	-	47° (range: 17–65°)
TH scoliosis at last follow-up	-	18.4° (range: 4–38°)
Mean TH scoliosis improvement	0.9°	10.7°
TL/L scoliosis pre-brace	25.1° (range: 13–40°)	30.8° (range:18–40°)
TL/L scoliosis in 1st brace	16.3° (range: 0–34°)	22.8° (range: 14–32°)
TL/L scoliosis at end of bracing	21° (range: 0–38°)	40° (range: 22–66°)
TL/L scoliosis preoperatively	-	44.3° (range: 28–66°)
TL/L scoliosis at last follow-up	-	17.2° (range: 2–45°)
Mean TL/L scoliosis improvement	4.1°	13.6°
Triradiate cartilage	Open: 27 Closed: 44	Open: 15 Closed: 11
Risser grade (start of bracing)	0.4 (range: 0–2)	0.19 (range: 0–2)
Risser grade (end of bracing)	4.7 (range: 4–5)	4.2 (range: 1–5)
Age (start of bracing); years	12.9 (range: 9.9–15.9)	12.6 (range: 10.3–15.8)
Age (end of bracing); years	16 (range: 14.8–18.4)	15 (range:11.9–17.6)
Age at scoliosis surgery; years	-	15.4 (range: 12.6–17.8)
Years in brace treatment	3.2 (range: 0.5–9.1)	2.1 (range: 0.2–4.2)
Follow-up (after bracing); years	4 (range: 0.9–8.7)	-
Follow-up (post-surgery); years	-	5 (range: 1.8–8.9)

TH: thoracic; TL: thoracolumbar; L: lumbar.

**Table 5 healthcare-11-01491-t005:** Patients with non-IS who completed brace treatment and underwent scoliosis surgery.

Data	Congenital Scoliosis (9 Patients)	NM Scoliosis (3 Patients)	Syndromic Scoliosis (21 Patients)	Secondary Scoliosis (14 Patients)
Type of scoliosis	TH: 2; TH and L: 7	TH and L: 2; TL/L: 1	TH: 8; TH and L: 9; double TH: 1; TL/L: 3	TH: 6; TH and L: 7; TL/L: 1
TH scoliosis pre-brace	40.4° (range: 25–59°)	32° (range: 27–37°)	35.8° (range: 20–65°)	47.2° (range: 21–78°)
TH scoliosis in 1st brace	32.3° (range: 15–46°)	26.5° (range: 24–29°)	28.2° (range: 15–54°)	33.9° (range: 14–51°)
TH scoliosis at end of bracing	50.1° (range: 10–72°)	53° (range: 46–60°)	72° (range: 43–98°)	74° (range: 34–117°)
TH scoliosis preoperatively	56.1° (range: 10–72°)	56° (range: 52–60°)	76.5° (range: 50–98°)	74° (range: 34–117°)
TH scoliosis at last follow-up	32.6° (range: 11–43°)	29.5° (range: 28–31°)	31.8° (range: 17–45°)	31.7° (range: 17–63°)
Mean TH scoliosis improvement	7.8°	2.5°	4°	15.5°
TL/L scoliosis pre-brace	32.1° (range: 22–48°)	35.5° (range: 31–40°)	33.5° (range: 14–58°)	41.8° (range: 24–51°)
TL/L scoliosis in 1st brace	24.2° (range: 16–33°)	21° (range: 17–23°)	20.8° (range: 10–35°)	25.5° (range: 18–35°)
TL/L scoliosis at end of bracing	45.5° (range: 13–85°)	57.8° (range: 36–67°)	57.7° (range: 39–81°)	38.4° (range: 27–70°)
TL/L scoliosis preoperatively	48.3° (range: 13–85°)	60.3° (range: 46–67°)	60° (range: 43–81°)	48.4° (range: 38–70°)
TL/L scoliosis at last follow-up	22.7° (range: 8–50°)	26.3° (range: 18–32°)	22.9° (range: 8–30°)	17.5° (range: 12–31°)
Mean TL/L scoliosis improvement	9.4°	9.2°	10.6°	24.3°
Triradiate cartilage	Open: 9	Open: 3	Open: 21	Open: 13; Closed: 1
Risser grade (start of bracing)	0.5 (range: 0–1)	0	0	0
Risser grade (end of bracing)	4.5 (range: 4–5)	4.5 (range: 4–5)	2.3 (range: 0–5)	3.6 (range: 0–5)
Age (start of bracing); years	5.3 (range: 1.6–11.7)	7.2 (range: 6.2–8)	6.8 (range: 2.2–14.2)	8.8 (range: 1.7–3.6)
Age (end of bracing); years	12.9 (range: 10.2–15.8)	13.7 (range: 12.7–15.1)	12.4 (range: 9.6–16.7)	12.6 (range: 9.9–15.9)
Age at scoliosis surgery; years	13.3 (range: 10.6–16.1)	13.9 (range: 12.7–15.6)	12.7 (range: 10.4–16.7)	12.6 (range: 10.3–5.9)
Years in brace treatment	5.2 (range: 2.1–8.8)	6.7 (range: 5.3–7.7)	5.8 (range: 2.4–11.8)	4 (range: 1.2–7.3)
Follow-up (post-surgery); years	4.5 (2.5–5)	4.5 (2–4.8)	4.1 (2.4–4.5)	4.2 (2.6–5)

TH: thoracic; TL: thoracolumbar; L: lumbar.

**Table 6 healthcare-11-01491-t006:** Multiple logistic regression models of factors associated with the success of brace treatment.

Characteristics	Odds Ratio	95% Confidence Interval	*p*-Value
Scoliosis angle	1.684	1.153–2.657	0.03 *
Triradiate cartilage	1.719	1.165–3.475	0.02 *
Risser grade	1.518	0.872–2.914	0.04 *
Menarche	1.894	1.243–3.576	0.03 *
Idiopathic diagnosis	3.982	2.771–4.956	0.008 *
AIS	2.365	1.917–3.867	0.01 *

* Statistically significant if *p* < 0.05

**Table 7 healthcare-11-01491-t007:** Brace questionnaire results (patients with IS and non-IS who completed brace treatment and did not require scoliosis surgery—109/118 patients could complete the questionnaire).

	Mean Score	Range	Yes	No
Did you understand the purpose of brace treatment?	-		109	0
2. Did you stop wearing the brace sooner than anticipated?			9	100
3. During treatment, how satisfied were you with the information provided and advice given about wearing the brace? (0 = very dissatisfied; 5 = very satisfied)	4.8	3–5	-	-
4. During treatment, I felt I was well supported by my Orthotist. (0 = strongly disagree; 5 = strongly agree)	4.9	3–5	-	-
5. I felt any brace-related problems were quickly attended to by my Orthotist. (0 = strongly disagree; 5 = strongly agree)	4.9	4–5	-	-
6. How satisfied were you with the ease of contacting your Orthotist? (0 = very dissatisfied; 5 = very satisfied)	4.8	3–5	-	-
7. How satisfied were you with the quality of your brace? (0 = very dissatisfied; 5 = very satisfied)	4.6	3–5	-	-
8. How much did wearing the brace affect your day-to-day activities? (0 = not at all; 5 = very much)	3.1	0–5	-	-
9. How much did wearing the brace affect your sleeping pattern? (0 = not at all; 5 = very much)	2.0	0–5	-	-
10. Looking back, do you feel brace treatment was worth it? (0 = not at all; 5 = very much)	4.6	1–5	-	-
11. How likely would you be to recommend brace treatment to a patient with a similar condition? (0 = very unlikely; 5 = very likely)	4.5	3–5	-	-

## Data Availability

The data supporting our reults is available in the database of our National Service. Data on our patients is submitted in the British Spine Registry and presented in detail in half-year and annual reports to the National Services Division of the Scottish Government who is the Commissioner of our National Service.

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
