# Peer review of "Effectiveness of the Boston Brace in the Treatment of Paediatric Scoliosis: A Longitudinal Study from 2010–2020 in a National Spinal Centre"

_healthcare, 2023, doi:10.3390/healthcare11101491_

Round 1

Reviewer 1 Report

Tsirikos et al. surveyed the effectiveness of Boston brace in the pediatric scoliosis cases, in term of degrees of reductions in the spinal rotation, requirement of surgical corrections, etc, between the year of 2010 and 2020 in this “National Service”.  The study is overall meaningful, with the data presented supportive of the conclusions the authors have made.  However, the language can and need to be improved.  The overall language style appears to be too casual for the scientific writing.  I recommend the authors hire a professional scientific editor to help improve the language.  Additionally, there are these corrections or clarifications I recommend, as detailed below:

1.       Please complete the authors’ affiliations.  The numerical references next to the authors’ names suggest the authors are affiliated with five organizations; however, there is only one listed.

2.       Through the manuscript, the authors used a symbol which looks like a low case letter “o” to refer to the degrees of spine rotation.  Please use the symbol “ ° ” to refer to degrees.

3.       In Line 54 and Line 57, the authors stated critical facts without citing articles.  Please add these necessary references.

4.       In Line 61, and a few other places in the manuscript, the authors refer to certain clinical organization as “our Service”, similar to “a National Service” in the title.  I find this saying confusing.  Is it a clinic, hospital or treatment center?  Please clarify.

5.       A long and ambiguous sentence expands from Line 83 to Line 86, with a similar case in Line 89 – 92, as well as Line 93-96.  Please reconstruct these sentences.

6.       In the paragraph in Line 129-141, the authors mixed the usage of Arabic numbers and spelling these numbers out in English.  Please unify the format to Arabic numbers.

7.       In Line 143-145, the authors kept inserting the article “an” in front of the abbreviation of a condition, IS.  Please eliminate the word “an”, if IS here refers to idiopathic scoliosis.

8.       Similar to above, in the paragraph in Line 204-214, the authors mixed the usage of Arabic numbers and spelling these numbers out in English.  Please unify the format to Arabic numbers.

9.       The data tables are out of order, with Table 4 on Page 4, after Table 1 and before Table 2.

1.   Please include the p values the authors indicated in the text in all the data tables applicable.

Author Response

Dear Editor,

We would like to thank you and the Reviewers for taking the time to go through our paper entitled: ‘Effectiveness of the Boston brace in the treatment of paediatric scoliosis: a longitudinal study from 2010-2020 in a National Service’ and for your consideration for possible publication in the Special Issue of Healthcare on Paediatric Spinal Deformity.

In response to the Reviewers’ comments we made the following changes:

Reviewer 1

Tsirikos et al. surveyed the effectiveness of Boston brace in the pediatric scoliosis cases, in term of degrees of reductions in the spinal rotation, requirement of surgical corrections, etc, between the year of 2010 and 2020 in this “National Service”.  The study is overall meaningful, with the data presented supportive of the conclusions the authors have made.  

Answer: Thank you very much for these positive comments.

However, the language can and need to be improved.  The overall language style appears to be too casual for the scientific writing.  I recommend the authors hire a professional scientific editor to help improve the language.  

Answer: The manuscript has been reviewed by the Medical Writers’ Service which is available in the Medical School of the University of Edinburgh. Changes throughout the text have been made in the revised manuscript and the advice received by the Medical Writers is that the revised paper is appropriate for publication in a Medical Journal.

Additionally, there are these corrections or clarifications I recommend, as detailed below:

  1. Please complete the authors’ affiliations.  The numerical references next to the authors’ names suggest the authors are affiliated with five organizations; however, there is only one listed.

Answer: The affiliation for all authors is the Scottish National Spine Deformity Centre, Royal Hospital for Children and Young People, Edinburgh, UK. This information was included in the original submission.

  1. Through the manuscript, the authors used a symbol which looks like a low case letter “o” to refer to the degrees of spine rotation.  Please use the symbol “ ° ” to refer to degrees.

Answer: We used the symbol ‘o’ to refer to degrees throughout the manuscript.

  1. In Line 54 and Line 57, the authors stated critical facts without citing articles.  Please add these necessary references.

Answer: We have added relevant references as suggested by the Reviewer in the revised manuscript (references 8-12).

  1. In Line 61, and a few other places in the manuscript, the authors refer to certain clinical organization as “our Service”, similar to “a National Service” in the title.  I find this saying confusing.  Is it a clinic, hospital or treatment center?  Please clarify.

      Answer: We referred throughout the text to ‘a National Spinal Centre’ which reflects the Spinal Unit where the study was conducted (Scottish National Spine Deformity Centre).  

  1. A long and ambiguous sentence expands from Line 83 to Line 86, with a similar case in Line 89 – 92, as well as Line 93-96.  Please reconstruct these sentences.

      Answer: These sentences have now been revised as requested.

  1. In the paragraph in Line 129-141, the authors mixed the usage of Arabic numbers and spelling these numbers out in English.  Please unify the format to Arabic numbers.

      Answer: This point has now been rectified in the revised text.

  1. In Line 143-145, the authors kept inserting the article “an” in front of the abbreviation of a condition, IS.  Please eliminate the word “an”, if IS here refers to idiopathic scoliosis.

      Answer: This point has now been rectified throughout the text in the revised manuscript.

  1. Similar to above, in the paragraph in Line 204-214, the authors mixed the usage of Arabic numbers and spelling these numbers out in English.  Please unify the format to Arabic numbers.

     Answer: This point has now been rectified in the revised text.

  1. The data tables are out of order, with Table 4 on Page 4, after Table 1 and before Table 2.

      Answer: This is an editing mistake when the PDF was created following our submission as the Tables were submitted in the original paper in the correct order of appearance.

  1. Please include the p values the authors indicated in the text in all the data tables applicable.

Answer: The p values were calculated for the whole group of patients with idiopathic or non-idiopathic scoliosis accordingly. To have larger numbers for doing the paired t-test we grouped all patients with IS or non-IS who had a thoracic or a thoracolumbar/lumbar scoliosis and calculated the p values. This is the reason why the p values are only included within the text in the section under ‘Results’. The tables present the radiological data for the separate sub-groups of patients within the IS cohort as AIS, JIS and IIS (Tables 2-4). Similarly, Table 5 presents the radiographic parameters in the sub-groups of patients with non-IS (congenital, NM, syndromic and secondary scoliosis).   

We would like to thank you and the Reviewers in advance for taking the time to go through our paper and make your constructive comments.

We are looking forward to hearing from you soon.

Kind regards.

Yours sincerely,

The authors of this paper

Reviewer 2 Report

Authors are to be commended for tackling such a huge project as bracing results for scoliosis of varying etiology.  Unfortunately that makes it harder to control for all variables that might affect a full time bracing program, especially in the non IS group when actual brace wear time can be more difficult to monitor. Some comments:

1. Please edit the entire text with regard to the ‘degree’ symbol - it is entered as a ‘o’ letter and not as a symbol.

2. When describing curves types in the sub-groups, most readers now rely on Lenke classification rather than descriptors of the curve.

3. A common criticism of any bracing study is compliance - while the recommendation was for 20 hours a day, the manuscript only assumes bracing for that amount of time.  It also would be helpful to more clearly define what a brace failure was at the beginning of the materials/result section - was it 5 degrees or 10 degrees progression, or just less than 50 degrees of curve at cessation of bracing?

4. I found all the different types of scoliosis and subsequent results a bit confusing to follow and a detractor from understanding bracing results in AIS or IS in general.  My suggestion would be to focus on IS only for this paper.

5. Seems like the real study group is the 249 that completed brace treatment, ( and we are to assume similar results with the remaining 231 still in brace? ) - otherwise, I would stick to analyzing just the 249.

6. Unfortunately, 2 year follow up on a 48 degree curve that was “successfully” treated with a brace, may become a surgical candidate as a young adult.  Curve location also important since many 40 degree lumbar or TL curves do not hold up well into young adulthood.  This is why it may be difficult to define what is successful brace result - and why holding progression to only 5 degrees is the goal of bracing, while keeping curves at 35 degrees or less (in most instances).

Author Response

In response to the Reviewers’ comments we made the following changes:

Reviewer 2

Authors are to be commended for tackling such a huge project as bracing results for scoliosis of varying etiology.  Unfortunately that makes it harder to control for all variables that might affect a full time bracing program, especially in the non IS group when actual brace wear time can be more difficult to monitor. Some comments:

  1. Please edit the entire text with regard to the ‘degree’ symbol - it is entered as a ‘o’ letter and not as a symbol.

Answer: This is an editing mistake as the ‘degree’ symbol was entered as ‘o’ in the original submission throughout the text.

  1. When describing curves types in the sub-groups, most readers now rely on Lenke classification rather than descriptors of the curve.

Answer: We used the description of scoliotic curves according to their location apex as thoracic, thoracolumbar and lumbar due to the fact that the Lenke classification only applies to AIS but not in scoliosis related to other aetiologies.

  1. A common criticism of any bracing study is compliance - while the recommendation was for 20 hours a day, the manuscript only assumes bracing for that amount of time.  It also would be helpful to more clearly define what a brace failure was at the beginning of the materials/result section - was it 5 degrees or 10 degrees progression, or just less than 50 degrees of curve at cessation of bracing?

Answer: We totally agree with this comment by the Reviewer. Compliance is a challenging point when bracing is applied in young patients. We used the criteria for ‘treatment success’ or ‘treatment failure’ that were applied in the BRAIST which is the prime study that established the effectiveness of bracing in patients with AIS. This point is clarified within the section under ‘Material and Methods’.

In addition, we explained in the section under ‘Discussion’ that preventing scoliosis from reaching 50o at the end of spinal growth was the benchmark for effectiveness of brace treatment in our study similar to the BRAIST and in accordance with the SRS Guidelines (reference 17).    

  1. I found all the different types of scoliosis and subsequent results a bit confusing to follow and a detractor from understanding bracing results in AIS or IS in general.  My suggestion would be to focus on IS only for this paper.

Answer: There are several previous reports on the effectiveness of bracing in patients with AIS of 20-40o. We see as a major strength of our study the fact that we included groups of patients with different underlying aetiologies of scoliosis (non-IS) to assess whether brace treatment has a role in delaying or avoiding the need for scoliosis correction.

We also included patients with idiopathic scoliosis above 40o as this group reflects the type of patients that are commonly seen in a paediatric spinal practice and where bracing can prevent the need for repetitive surgery (such as growing rod procedures) that increases the risk of complications and major morbidity. 

  1. Seems like the real study group is the 249 that completed brace treatment, ( and we are to assume similar results with the remaining 231 still in brace? ) - otherwise, I would stick to analyzing just the 249.

Answer: We have included generic information in the total group of 480 patients who were part of our bracing programme. However, we analysed in this paper the 249 patients who completed bracing and had minimum 2-year follow-up beyond the end of brace treatment or after scoliosis surgery.

  1. Unfortunately, 2 year follow up on a 48 degree curve that was “successfully” treated with a brace, may become a surgical candidate as a young adult.  Curve location also important since many 40 degree lumbar or TL curves do not hold up well into young adulthood.  This is why it may be difficult to define what is successful brace result - and why holding progression to only 5 degrees is the goal of bracing, while keeping curves at 35 degrees or less (in most instances).

Answer: We agree with the point made by the Reviewer. We included this point within the limitations of our study in the section under ‘Discussion’. Unfortunately, this is a limitation of any other bracing study. To address this issue we are following our patients into adult life to document curve progression beyond skeletal maturity that may require surgical management.   

We would like to thank you and the Reviewers in advance for taking the time to go through our paper and make your constructive comments.

We are looking forward to hearing from you soon.

Kind regards.

Yours sincerely,

The authors of this paper

Reviewer 3 Report

The article starts well and is fairly easy to follow. 

The part devoted to statistical analysis lacks rigour and needs some substantial adjustments.

1) In the "Multiple logistic regression analysis" section, the estimated model is not clear. If I understand correctly, the outcome in the model is "success" in healing with the brace. However, in the linear predictor, I do not see the variable that indicates whether a patient used a brace or not. It is impossible to assess the effectiveness of the method if it is not included in the model and if there are no subjects among the patients who did not wear the brace. 

2) I would carefully check the p-values in Table 6. Is the one for "Risser Grade" less than 0.05?

3) The SRS-22r values are not numerical, they are a scale. Therefore, it is incorrect to average the values and to perform t-tests. Instead, it would be correct to assess the medians using appropriate tests (e.g. Wilcoxon-Mann-Whitney test). 

There are also some other minor comments:

4) Are the authors' affiliations missing or are they all from the same affiliation? 

5) Section 3 (Results): why do you sometimes write patient numbers in letters and others in numbers? 

Author Response

Reviewer 3

The article starts well and is fairly easy to follow. 

The part devoted to statistical analysis lacks rigour and needs some substantial adjustments.

  • In the "Multiple logistic regression analysis" section, the estimated model is not clear. If I understand correctly, the outcome in the model is "success" in healing with the brace. However, in the linear predictor, I do not see the variable that indicates whether a patient used a brace or not. It is impossible to assess the effectiveness of the method if it is not included in the model and if there are no subjects among the patients who did not wear the brace. 

Answer: We compared in our analysis patients who were braced and did not reach 50o at the end of their spinal growth (success of bracing) to those who were braced but the scoliosis exceeded 50o and required surgical correction (failure of bracing). This is the same approach as that used by Ovadia et al. in a paper published in the Journal of Child Orthopaedics in 2012 (reference 20 in our revised manuscript). We included this reference in our section under ‘Discussion’ as a direct comparison to our study. We also included two previous studies that reported on prognostic factors that could predict the effectiveness of bracing (references 18 and 19).    

  • I would carefully check the p-values in Table 6. Is the one for "Risser Grade" less than 0.05?

Answer: The p values in Table 6 are correct. However, the OR for Risser grade is 1.518. This has been corrected in the revised manuscript.

  • The SRS-22r values are not numerical, they are a scale. Therefore, it is incorrect to average the values and to perform t-tests. Instead, it would be correct to assess the medians using appropriate tests (e.g. Wilcoxon-Mann-Whitney test). 

Answer: We used the t-test to compare the SRS-22r scores (total and individual domains) for those patients who completed bracing and underwent scoliosis surgery. These patients had SRS-22r scores available at both points of assessment. We did not average the values before performing t-tests. However, at the Reviewer’s recommendation we removed the p values from the section under ‘Results’ that presents the ‘Patient reported outcomes’ in the revised manuscript. 

There are also some other minor comments:

  • Are the authors' affiliations missing or are they all from the same affiliation? 

           Answer: All authors have the same affiliation.

  • Section 3 (Results): why do you sometimes write patient numbers in letters and others in numbers? 

          Answer: All patient numbers are presented in Arabic numbers within the revised text.

All changes in the revised manuscript are in bold letters and underlined.

-All authors have approved the revised manuscript and agree with its submission to Healthcare.

-We confirm that neither the manuscript nor any parts of its content are currently under consideration or published in another journal.

We would like to thank you and the Reviewers in advance for taking the time to go through our paper and make your constructive comments.

We are looking forward to hearing from you soon.

Kind regards.

Yours sincerely,

The authors of this paper